# A Circular Biorefinery-Integrating Wastewater Treatment with the Generation of an Energy Precursor and an Organic Fertilizer

**Tabassum-Abbasi [1,*], Pratiksha Patnaik [2], Ranjan Rahi [3] and Shahid Abbas Abbasi [2]**

1   School of Engineering, University of Petroleum and Energy Studied, Dehradun 248 007, India
2   Centre for Pollution Control & Environmental Engineering, Pondicherry University, Puducherry 605 014, India; pratikshapatnaik@rediffmail.com (P.P.); abbasi.cpee@gmail.com (S.A.A.)
3   Society for Innovation & Entrepreneurship, Indian Institute of Technology, Emergy Enviro Private Limited, Mumbai 400 076, India; ranjan.rahi87@gmail.com
*   Correspondence: tab.abbasi@gmail.com

**Abstract:** A circular (close-loop) biorefinery, which integrates wastewater treatment with the generation of an energy precursor and organic fertilizer, tested at the level of a pilot plant treating 54,000 L per day (LPD) of sewage, is described. In the biorefinery's first stage, sewage was treated in a novel SHEFROL® (sheet-flow-root-level) bioreactor at a very rapid rate, indicated by a hydraulic retention time of a mere 6 h, to a level that met the prevailing national standards for the discharge of treated sewage. The main bioagent of the reactor—water hyacinth—was then processed for the generation of energy precursors. For this, volatile fatty acids (VFA) were extracted in a simple batch reactor operating at ambient temperature and pressure. The 'spent' weeds were then converted into organic fertilizer, also at ambient temperature and pressure, by the high-rate vermicomposting process earlier reported by the authors. In this manner, wastewater treatment, energy production, and the generation of a fertilizer were achieved rapidly and efficiently, creating a circular close-loop system that required very little energy and materials and generated almost zero net waste.

**Keywords:** biorefinery; sewage treatment; energy precursors; organic fertilizer; closed loop



## 1. Introduction

The implementation of the biorefinery concept has so far been predominantly confined to the use of algae as the main bioagent [1,2]. A quick assessment of 75 randomly picked research publications on wastewater biorefineries reveals that as many as 95% of the reports are centred around algae. In contrast, the exploration of the use of vascular plants has been very limited, confined so far to duckweed [3,4], and *Pistia stratiotes* [5,6]. In a conceptual review, Nawaj-Alam et al. [7] dwelt, at length, on the potential of aquatic weeds as candidates for use in biorefineries. Among the attributes of aquatic weeds that Nawaj-Alam et al. [7] identified as highly relevant to biorefineries are the weeds': (a) high growth rate; (b) resilience and robustness; (c) proven ability to phytoremediate a large variety of pollutants; (d) capacity to provide a very wide range of chemicals; (e) potential use as feedstock in the making of fertilizers, paper pulp, alcohol, biodiesel, etc.; and (f) use as a source of clean energy in the form of biomethane and biohydrogen.

Additionally, the use of vascular plants in the wastewater treatment step of the biorefineries does not necessitate the use of the agitation and filtering devices that are necessary in algae-based biorefineries. Further, due to their high productivity, vascular plants capture solar energy much more efficiently than corresponding masses of algae do [8,9].

Given the above-mentioned attributes of aquatic weeds, especially their well-established ability to purify sewage and phytoremediate other forms of polluted water [10,11], greater attention appears warranted on the development of biorefineries that revolve around the use of aquatic weeds in particular, and weeds in general.

The first report on exploring an aquatic weed—pistia (*Pistia stratiotes*)—in a biorefinery context describes the assessment of the biomass production of *P. stratiotes* while the weed was used in phytoremediation [5]. In simulated experiments, *P. stratiotes* was grown on synthetic wastewater (SW) and on polluted river water (PRW), with or without augmenting the PRW with fertilizers. Year-round measurements showed a high rate of biomass production that varied with seasons, while remaining significant throughout. The work showed the feasibility of wastewater treatment allied with high-rate biomass generation. The authors subsequently extended the work to a 13,000 L phytoremediation lagoon [6] and found that, within a hydraulic retention time of 7 days, polluted river water was substantially treated in terms of the removal of COD, ammonical nitrogen, nitrate nitrogen, and phosphorus to the extents of 48–88%, 77–99%, 17–97%, and 74–93%. The average daily weed productivity was 58.1 Kg/hectare.d. However, the authors did not shed light on what use the *P. stratiotes* biomass could be put to. This is a very important question because *P. stratiotes* is one of the most invasive and colonizing of all weeds and huge quantities of its biomass is generated in natural water bodies [11]. Hence, the availability of *P. stratiotes* biomass is not a constraint, but finding an economically viable means of its utilization is [11].

Calicioglu et al. [3] investigated the duckweed (*lamnacae*) utilization step of a hypothetical wastewater-treatment-cum-resource-recovery biorefinery to show that, by the sequential integration of two or three of the ethanol fermentation, volatile fatty acids generation, and methane production steps, up to $0.69 \pm 0.07$ g of a carbon-containing by product yield can be obtained per gram of duckweed carbon. Later, the same authors [4] reported a techno-economic analysis accompanied by a life-cycle assessment of a hypothetical wastewater-derived duckweed biorefinery but no experiment-based validation was reported, much less any large-scale trials.

The foregoing assessment of the state-of-the-art technologies reveals that, even though the use of aquatic weeds in particular and weeds in general carry great promise of driving successful biorefineries, this aspect has remained largely unexplored so far. The lone previous effort described earlier, in which a reasonably large system—a 13,000 L experimental lagoon—was used, was by Olguin et al. [6]. In it, the significant phytoremediation of polluted river water required an HRT of 7 days. Any system running at such a high HRT would need significantly large land areas. This constraint, of high HRT and consequently large land-area requirements, has been the main reason for limiting the use of constructed wetlands in wastewater treatment [10,12]. The techno-economic and life-cycle assessment of Calicioglu et al. [4] also showed that duckweed-pond construction can lead to increased land-use change impacts, implying that the greater the HRT and consequently greater the land-use, the bigger the negative impacts.

The present work is arguably the first ever in which a life-size wastewater-driven biorefinery of 54,000 L per day (LPD) sewage-treatment capacity, is described that is based on an aquatic weed—water hyacinth (*Eichhornia crassipes*). The sewage-treatment step was handled by the recently developed, patented, and trade-marked SHEFROL® (sheet-flow-root-level) bioreactor. It operated at an HRT of merely 6 h, which represents as fast a rate of treatment as achievable by the activated-sludge process and its variants [13,14]. It also represents a rate that is orders of magnitude faster than the rate achieved by Olguin et al. [6] with the 7-day HRT of their phytofilteration lagoon.

The wastewater treatment accompanied weed growth was periodically harvested to, first, extract from it energy precursors in the form of volatile fatty acids (VFAs) and, then, to convert the spent biomass into an organic fertilizer by deploying the process of high-rate vermicomposting recently developed by the authors [15]. In this manner, a closed-loop biorefinery was created that performs sewage treatment and yields two products in easy-to-fabricate and operate energy-frugal steps, while leaving nothing of which to dispose.

## 2. Materials and Methods

### 2.1. Setting Up the Sewage-Treatment System

The SHEFROL® reactor was designed for treating a maximum of 54,000 LPD of sewage, comprising of a portion of wastewater flow coming from a cluster of buildings situated in the campus of Pondicherry University. The reactor comprised of 8 channels in series (Figure 1). Each of the channels was 15 m long, 0.5 m wide, and 0.4 m deep. A flow equalization-cum-sedimentation tank of dimensions 3.5 m (length), 2 m (width), and 1.5 m (depth) was set upstream of the channels. Sewage was diverted to this tank using polyvinyl chloride (PVC) pipes, by gravity flow.

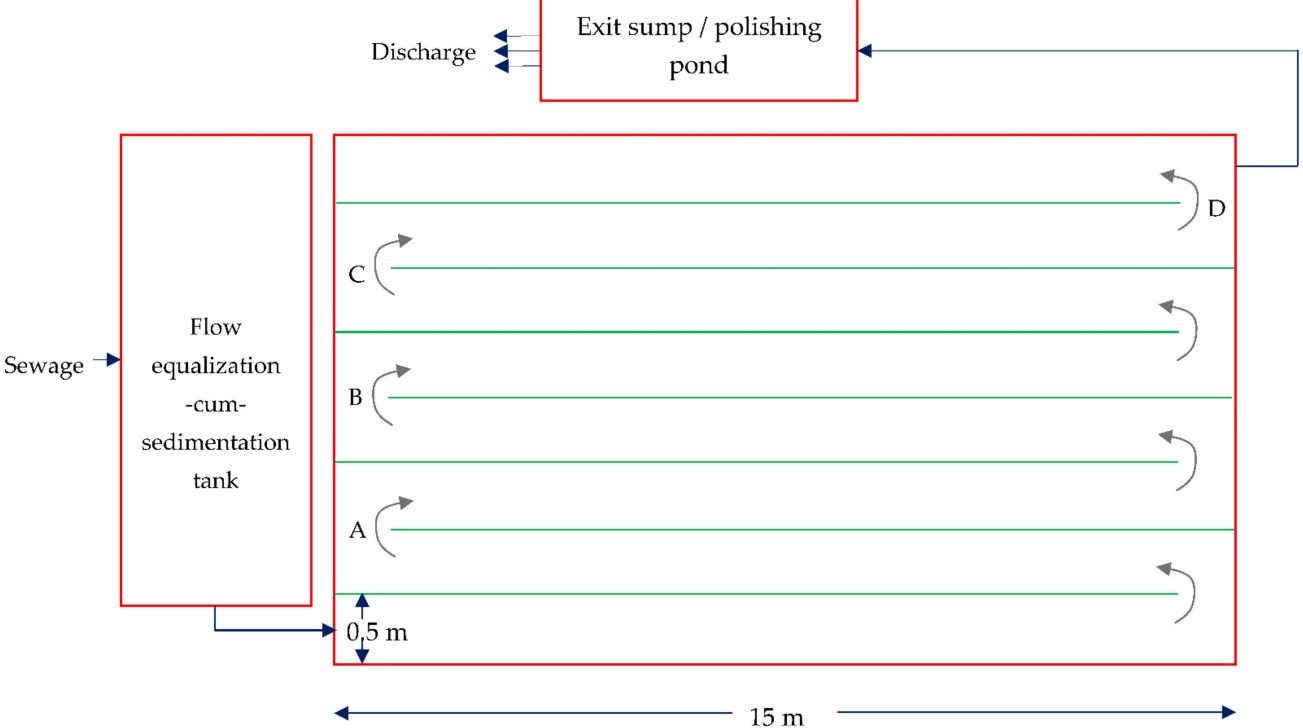

**Figure 1.** The schematic of the SHEFROL® unit. A, B, C, and D represent locations from where samples were drawn for HRTs of 2, 4, 6, and 8 h, respectively.

Assessment in a simulated SHEFROL® channel of 2 m width and 1.5 m depth had shown that introduction of *E. crassipes* in dense packing lifts the liquid level in the channel by 0.1 m. Allowing for this, the working volume of this SHEFROL® unit came to 18 m³.

To minimise the cost of construction of the reactor, discarded waste materials were used wherever possible, as detailed earlier when describing the commissioning of another SHEFROL® unit [16]. For the making of the below-ground feed sump and channels, soil was excavated according to their dimensions. They were supported with side walls made up of discarded cement bags in which the excavated soil had been filled. High density polyethylene (HDPE) sheets of 0.3 mm thickness were used as liners, to prevent percolation of the wastewater into the soil.

In the first phase of the experiments, effect of hydraulic retention time on the level of treatment, without and with macrophyte, was assessed. For this half of the design flow, 27,000 LPD was routed to the SHEFROL® unit to achieve a maximum hydraulic retention time of 8 h. Samples of the incoming raw wastewater and the outgoing effluent were drawn every day, at 14.00 h. Samples of sewage were also drawn from four points in its path, as marked in Figure 1, corresponding to hydraulic retention times (HRTs) of 2, 4, 6, and 8 h. The samples were preserved and analysed as per standard methods prescribed by the consortium of American Public Health Association and American Water Works

Association [17]. The extent of primary treatment was assessed in terms of removal of suspended solids (SS), the secondary treatment in terms of removal of chemical oxygen demand (COD), and tertiary treatment as represented by the removal of total Kjeldehl nitrogen (TKN) and soluble phosphorous (SP).

After the studies on the treatment achieved without planting the weed had been carried out for two weeks, all the channels were seeded with healthy, adult plants of *E. crassipes* taken from natural growths near the authors' workplace. The plants began to multiply rapidly and covered all channels from end to end within a week. The effect of HRT on the treatment achieved was then assessed in the manner described with the control experiment. At the completion of these experiments, the full design flow of 54,000 LPD was directed to the SHEFROL, thus carrying out sewage treatment at an HRT of 4 h.

### 2.2. Extraction of Energy Precursors from the Harvested Weed

Due to the abundance of nutrients in the wastewater that passes through the weed's roots, there is brisk and vigorous biomass production in the SHEFROL® channels. In addition, a fraction of plants keep suffering senescence upon completing their life span. Hence, biomass continues to be generated in SHEFROL® channels even as wastewater treatment is occurring there. The dead plants, together with overgrowth, were periodically harvested and put to downstream processing in the biorefinery.

For the first step of utilisation, harvested plants were air dried in the sun and subjected to acid-phase digestion for obtaining volatile fatty acids (VFAs). The VFAs are energy precursors because they can be converted into biogas in any functioning anaerobic digester [18]. The dried *E. crassipes* was powdered and pre-treated with aqueous NaOH solutions to break the weed's lignin matrix and enhance its biodegradability. It was then fed to acid phase reactors consisting of 15 L fibreglass vessels with provision of periodic stirring and product removal (Figure 2). Fresh cow dung, which is rich in microflora including cellulolytic, acidogenic, and acidogenic bacteria, was used as inoculum while the operating conditions were kept such that anaerobic conditions did not develop and the possibility of VFAs becoming decomposed into biogas in the acid phase reactors did not arise. The process parameters were optimised with the help of Taguchi method, which is considered to be a well-tried and tested procedure for process optimization [19,20]. For it, the Taguchi L9 matrix was set for four parameters at three levels: aqueous NaOH (2.5%, 5% and 7.5% weight/volume), soaking time (24, 48, and 72 h), NaOH reused (un-reused, once-reused, twice-reused), and cow dung concentrations (0.5%, 1% and 2.5%, dry weight basis). Experiments in duplicates were performed accordingly (Table 1). Measured quantities of powdered weed were pre-treated with aqueous NaOH, the pH of the mixture was adjusted to $7 \pm 0.5$, and the mixture transferred to the acid phase reactor (Figure 2). Dilution water and fresh cow-dung inoculum, quantified on the basis of dry weight earlier computed by drying a known quantity of cow dung to a constant weight at 105 C, were added. The contents were manually stirred for 30 s once every 6 h.

The analysis of the generated VFAs was carried out as per the procedure of distillation cum titration given in the Standard Methods for the Examination of Water and Wastewater [17]. The VFA production was monitored every day from the start of the reactor.

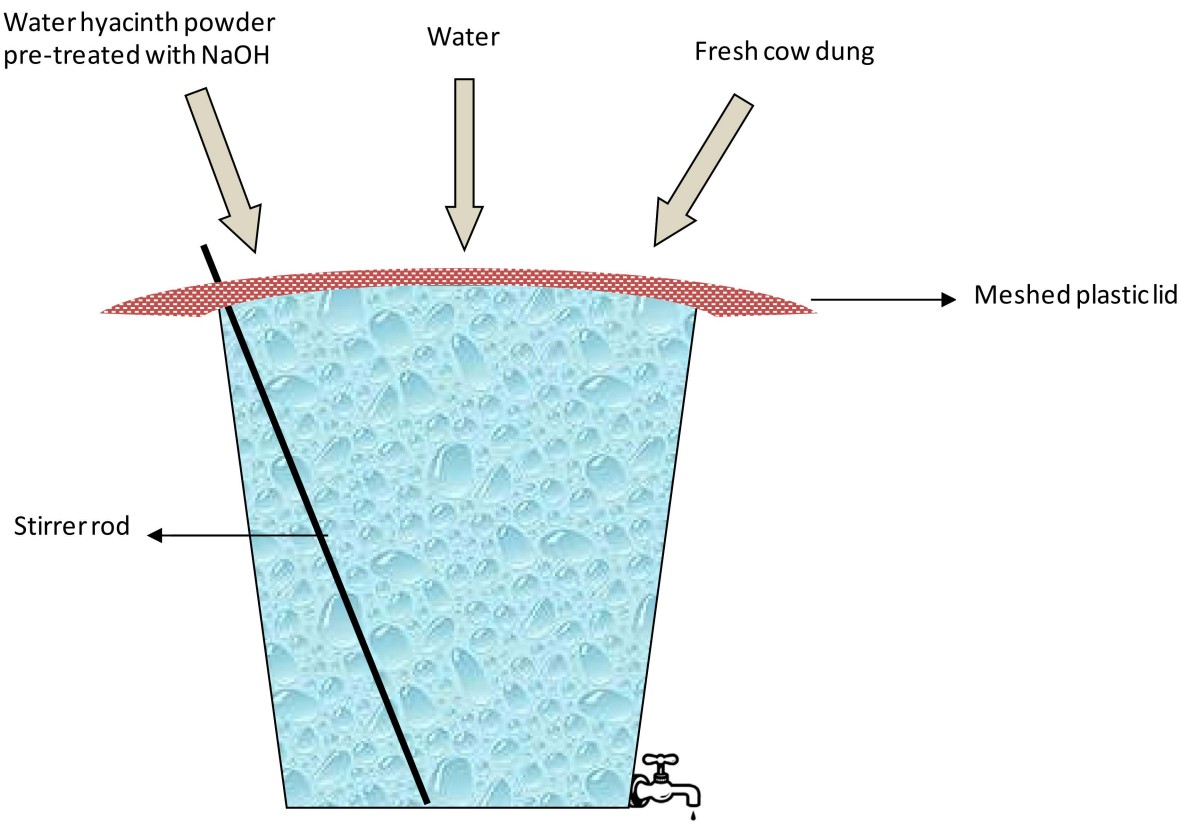

**Figure 2.** The VFA Reactor.

**Table 1.** The Taguchi L 9 matrix for optimizing process parameters, and the VFAs obtained from different combinations. The numbers in parenthesis in the penultimate column indicate the corresponding mean achieved by that row's process variables in the Taguchi plots presented later in this paper.

| Sodium Hydroxide Solution, g/100 mL | Duration, h, of Soaking the Weed Powder in NaOH | Cow Dung, g/100 mL | Reuse of NaOH Solution | VFA Obtained, g/Kg |
|---|---|---|---|---|
| 2.5 (1) | 24 (1) | 0 (1) | Fresh (1) | $39 \pm 2$ |
| 2.5 (1) | 48 (2) | 1 (2) | Reused once (2) | $51 \pm 3$ |
| 2.5 (1) | 72 (3) | 2.5 (3) | Reused twice (3) | $11 \pm 2$ |
| 5.0 (2) | 24 (1) | 1 (2) | Reused twice (3) | $50 \pm 3$ |
| 5.0 (2) | 48 (2) | 2.5 (3) | Fresh (1) | $73 \pm 4$ |
| 5.0 (2) | 72 (3) | 0 (1) | Reused once (2) | $45 \pm 2$ |
| 7.5 (3) | 24 (1) | 2.5 (3) | Reused once (2) | $49 \pm 3$ |
| 7.5 (3) | 48 (2) | 0 (1) | Reused twice (3) | $44 \pm 4$ |
| 7.5 (3) | 72 (3) | 1 (2) | Fresh (1) | $67 \pm 3$ |

*2.3. Generation of Organic Fertilizers from Spent Weed*

VFAs consist of carbon, oxygen, and hydrogen. Their removal from the weed leaves the weed's content of nitrogen, phosphorus and most other nutrients undiminished. On the other hand, soaking in water for several days, with or without NaOH, renders the weed soft and fragile. This aspect makes the spent weed an ideal feed for high-rate vermireactors developed earlier by S.A. Abbasi and co-workers [11,21]. Accordingly, the use of three species of earthworms—the anecic or geophytophagous *Dravida willsi* and the epigeics or phytophagous *Endriluseugeniae* and *Eisenia fetida*—was explored. The experimental details were essentially as reported recently for the high-rate vermicomposting of *Ipomoea carnia* [11].

## 3. Results and Discussion

### 3.1. Performance of the Sewage Treatment System

Discernable treatment of greywater began from the first day of the start of the reactor, as illustrated with the example of the removal of COD at the HRT of 6 h, in Figure 3. The extent of COD removal rose almost linearly with time to reach a peak by about the 10th day, achieving a steady state. Thereafter, the COD removal indefinitely hovered in the $73 \pm 3\%$ range. It remained steady in spite of much wider variations in the influent COD (91–238 mg/L; Table 1). The trend, in respect of all other parameters at all the four HRTs, was similar. In contrast, the COD removal in the control channel was only $9 \pm 2\%$, caused by natural aeration and sunlight as the greywater coursed through an unplanted SHEFROL® channel. The concentrations in the influent and their reduction, seen at four HRTs, in the five parameters studied by us is shown in Table 2. These five parameters were chosen as indicators of the primary treatment (exemplified by SS removal), secondary treatment (exemplified by the removal of COD and BOD), and tertiary treatment (as mirrored in the removal of TKN and SP) of the greywater. The results indicate that very substantial treatment occurred even at an HRT of just 2 h. The level of treatment increased significantly at the HRT of 4 h. Still-better treatment occurred at 6 h HRT. A further increase in the HRT brought only marginal improvement. Hence, an HRT of 6 h appears optimal because further slowing down the rate of throughput (by increasing the HRT) does not yield a corresponding improvement in the extent of treatment.

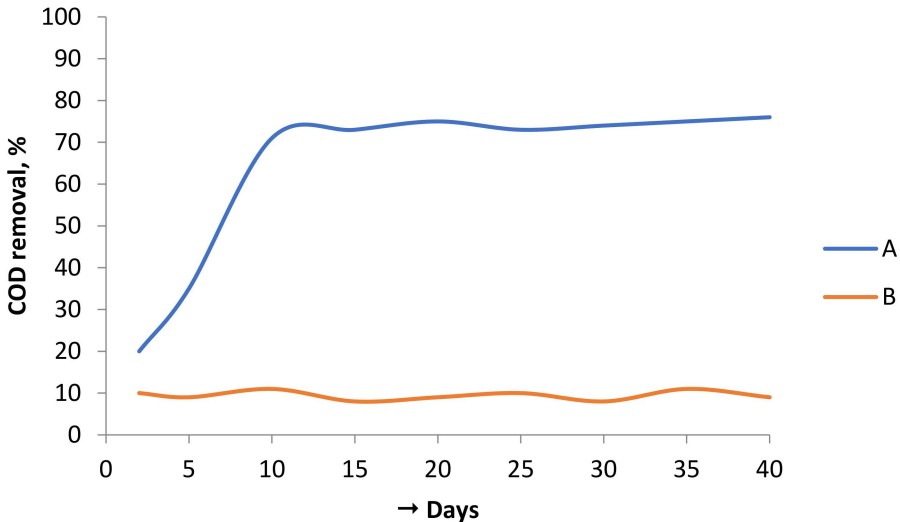

**Figure 3.** Pattern of COD removal as a function of time. Curve A: removal in channels planted with *E. crassipes*; curve B: removal in control (plant-free) channels.

**Table 2.** Levels of treatment achieved at different hydraulic retention times in the pilot-scale SHEFROL®.

| Parameter | Concentration in the Influent Greywater mg/L | Treatment, %, Achieved at HRT | | | |
|---|---|---|---|---|---|
| | | **2 h** | **4 h** | **6 h** | **8 h** |
| COD | 91–238 | 47–53 | 69–74 | 71–76 | 72–79 |
| BOD | 43–107 | 50–57 | 73–78 | 75–79 | 74–80 |
| SS | 48–121 | 39–44 | 70–75 | 75–87 | 80–91 |
| TKN | 23–44 | 27–33 | 29–36 | 35–41 | 35–44 |
| SP | 3–17 | 31–39 | 36–43 | 37–46 | 36–46 |

Considering that most conventional activated-sludge-based greywater treatment plants (ASPs) operate at HRTs of $\geq 6$ h, the rate of treatment achieved by the SHEFROL® unit is comparable in effectiveness with the ASPs. More significantly, SHEFROL® is seen to

achieve such a fast rate of treatment with much easier-to-fabricate-and-operate reactors. It also entails a great saving of electricity because of its total reliance on direct solar and gravitational energy. The latter is used in the form of exploiting the liquid head at the entry point of SHEFROL® to control the flow rate and HRT, thereby obviating the necessity of using a pump. Hence, SHEFROL® has a several times lower cost and ecological footprint than the ASPs.

### 3.2. Extraction of VFAs from Dead or Overgrown Water Hyacinth Plants Removed from SHEFROL®

Generation of significant quantities of VFAs occurred in the acid phase reactors (Figure 2) by the end of the first 24 h. The cumulative VFA production followed a bell-shaped curve as illustrated in Figure 4, reaching a peak in 4–6 days. The VFA concentration then began to decline—possibly because fresh VFA generation slowed down while the already-formed VFA began to decompose. Hence, an HRT of 5 days appears ideal for achieving maximum VFA yield. However, it is advisable to perform a few trial runs before setting up the HRT because the number of days to the peaking of the VFA yield may vary in the 4–8 day range, depending on ambient conditions. We have found it to be so in past studies [22,23].

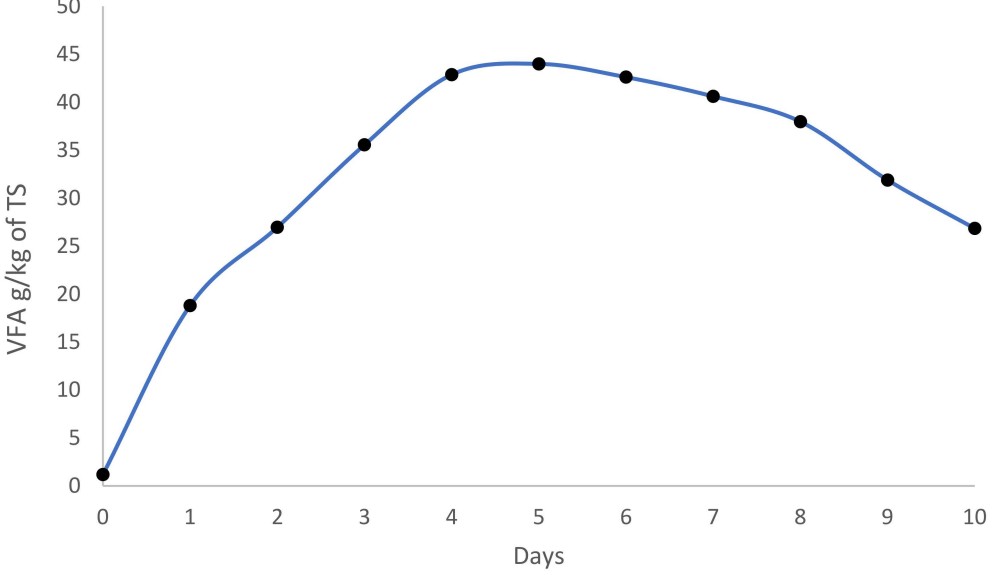

**Figure 4.** An illustrative example of the pattern of cumulative VFA production as a function of time in acid phase reactors.

The VFAs produced by different combinations of influencing parameters are given in Table 1. The main effects plots for means and SN (signal–noise) ratios, as generated on applying the Taguchi matrix, are as seen in Figures 5 and 6, respectively. Both types of plots have similar trends. It is seen that a change in the NaOH concentration from 2.5% to 5% has a very pronounced effect on VFA generation. A further increase to 7.5% causes a lowering in the VFA yield. Changes in soaking time and cow-dung concentration show a similar trend; the only difference being that a change from level 2 to level 3 brings a much sharper decline in the VFA yield in these two cases than was witnessed in case of NaOH concentration.

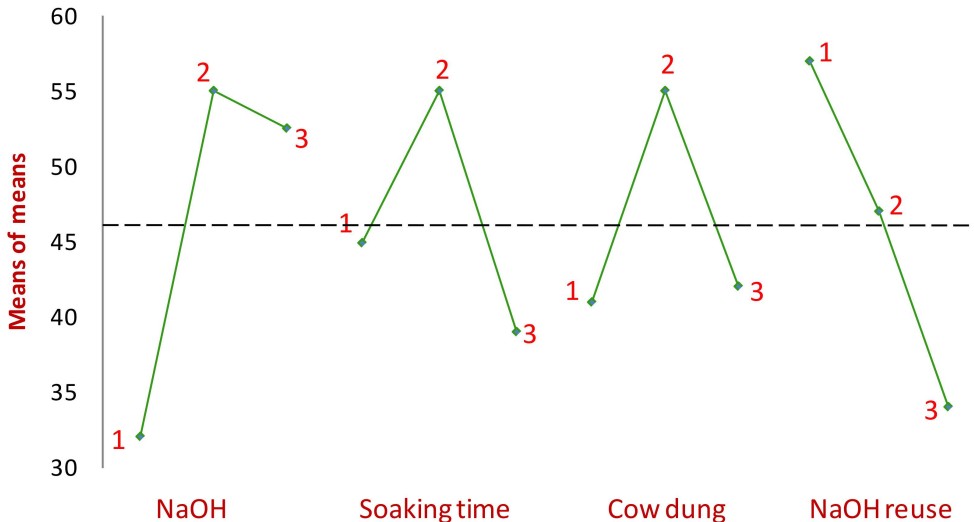

**Figure 5.** Main effects Taguchi plot for data means.

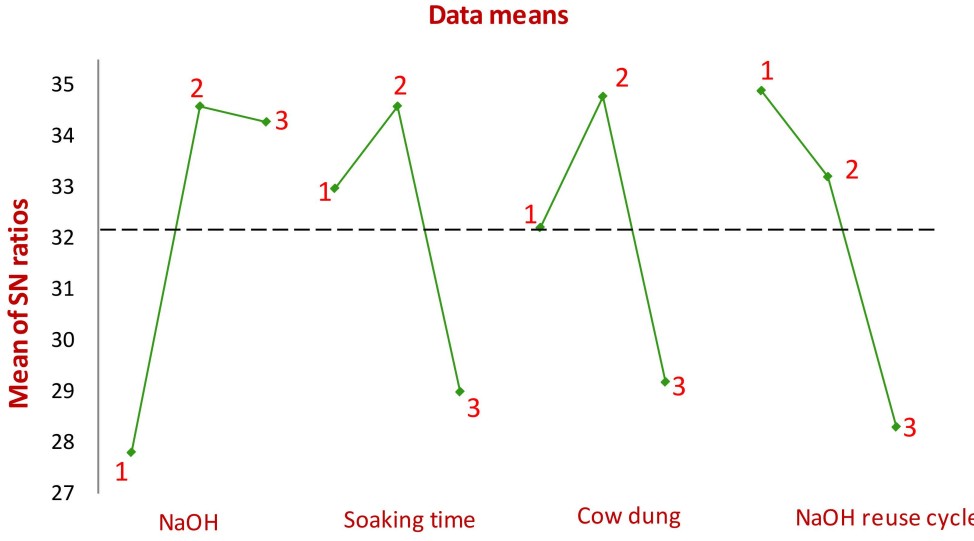

**Figure 6.** Main effects Taguchi plots for SN (signal–noise) ratios.

It is evident that the concentration of NaOH used in pre-treatment has a strong effect on the level of VFA generation, and 5% NaOH is deemed most preferable as it provides the maximum VFA yield. Further, a soaking time of 48 h and a cow-dung concentration of 1% appear optimal.

The reuse of NaOH solution exerted an unfavourable effect on the VFA yield. A fresh NaOH solution gave the best results, while the yield worsened with the number of times the NaOH solution was reused.

All-in-all, the Taguchi-matrix-based optimization indicated that, in order to maximize VFA yield, aqueous NaOH should be used at a 5 g/100 mL concentration. In it, the weed powder should be soaked for 48 h. A fresh NaOH solution should be used for pre-treatment each time and the concentration of cow-dung inoculum in the acid-phase reactor should be 1% (*w*/*v*).

### 3.3. Conversion of the Weed Biomass, Retrieved from Acid Phase Reactor, into Organic Fertilizer

As detailed in Abbasi et al. [11,21], the vermireactors were operated uninterruptedly in the 'pseudo-discretized continuous operation (PDCOP)' mode, wherein the effect of the natural biodegradation of the weed biomass, and of feed utilization by the earthworms born in the vermireactors, is minimized. This enables the realistic assessment of the course of vermicomposting as influenced by the adult earthworms with which the reactors were started.

The results of six months of uninterrupted vermireactor operation are summarized in Table 3. The first observation was taken 30 days after the start of the reactors because earthworms take about this much time to become acclimatized to the feed and also the vermireactor environment. All subsequent observations on vermicast production were taken once in 20 days, with the removal and quantification of vermicast, earthworm offshoots (juveniles and cocoons), and unutilised feed. Simultaneously, each reactor was restarted with fresh feed but with the same adult earthworms as were initially deployed. Hence, the data of Table 3 represents the pattern of vermicomposting achieved by the initial bunch of earthworms over 6 months, grazing upon fresh or nearly fresh feed all the time. All mass balance was performed on the basis of dry weights after the correlation between fresh weights and dry weights were determined for the feed, as well as the vermicompost, by oven drying known quantities to constant weight at 105 °C.

**Table 3.** Vermicast generated by the three species of earthworms over the course of six-month long continuous vermireactor operation.

| Number of Days from the Start | Vermicast Generated, mg, per Earthworm, per Day, by | | |
|:---:|:---:|:---:|:---:|
| | *E. eugeniae* | *E. fetida* | *D. willsi* |
| 30 | 26 ± 4 | 17 ± 3 | 11 ± 3 |
| 50 | 31 ± 5 | 22 ± 4 | 13 ± 3 |
| 70 | 32 ± 4 | 22 ± 5 | 13 ± 4 |
| 90 | 33 ± 4 | 24 ± 4 | 12 ± 4 |
| 110 | 32 ± 3 | 23 ± 3 | 13 ± 4 |
| 130 | 34 ± 5 | 23 ± 4 | 11 ± 3 |
| 150 | 31 ± 3 | 24 ± 4 | 12 ± 4 |
| 180 | 34 ± 3 | 22 ± 3 | 13 ± 2 |

The results have been presented in terms of vermicast generated per worm because that statistic can be used to directly estimate the number of earthworms that may be needed per unit mass of the feed to ensure it is quantitatively converted into vermicast at 20 days' solids retention time (SRT). Details on how the operation of the reactors in 'high-rate vermicomposting' mode enables near-complete conversion of the weed feed to vermicompost within 20 days while all conventional vermicomposting systems take 2–4 months to achieve the same results, was explained earlier [15].

The results reveal that, on a per animal basis, *E. eugeniae* provides the fastest rate of vermicast production from the spent weed *E. crassipes*, followed by *E. fetida*. The anacic *D. willsi* is, in comparison, significantly less efficient. However, if the assessment is based on earthworm size, hence, zoomass, *E. fetida* comes out to be on top because its average mass is 6 g while *E. eugeniae* is about twice as heavy and *D. willsi* about four times so. Hence, *E. fetida* should be preferred for this biorefinery, with *E. eugeniae* as the second choice. *D. willsi* appears too sluggish to be effective.

### 3.4. The Overall Process

With the steps described above, a fully closed-loop biorefinery can be operated, achieving wastewater treatment, energy production, and fertilizer production. All the steps can be taken at ambient temperatures and pressures, with the minimal input of materials and no net waste to dispose. All the three steps are also highly energy-frugal. It must be mentioned that there is significant natural variation in the sizes, attributes, and characteristics between

the same species of macrophytes growing in different regions. The agro-climatic conditions under which the macrophytes are used to treat sewage also vary significantly from location to location. In a similar way, intra-species variations exist among earthworms. Given these natural variations, the data generated in this paper should be treated as broadly indicative. It is advisable to fine tune new systems by performing test-runs before finalizing system designs.

## 4. Summary and Conclusions

The paper has described a close-loop biorefinery, which integrates wastewater treatment with the generation of energy and organic fertilizer. All the three steps occur in an energy-frugal, material-frugal, and inexpensive manner, with a negligible carbon footprint. First, sewage was treated on a pilot-plant scale of a 54,000 L per day capacity in a novel SHEFROL® (sheet-flow-root-level) bioreactor at a very rapid rate, indicated by a hydraulic retention time of a mere 6 h, to a level that met the prevailing national standards for the discharge of treated sewage [24]. The efficiency of the reactor, as indicated by the HRT, was comparable to the several times more expensive, activated-sludge-based processes currently in vogue. The main bioagent that was utilized in the reactor—water hyacinth—was then processed for the generation of energy precursors. For this, volatile fatty acids (VFAs), which can be fed to any anaerobic digester for obtaining fuel in the form of biogas, were extracted from the weed in a simple batch reactor operating at ambient temperature and pressure. The 'spent' weeds were then converted into organic fertilizer, also at ambient temperature and pressure, by the high-rate vermicomposting process earlier reported by the authors. In this manner, wastewater treatment, energy production, and generation of fertilizer were all achieved rapidly and efficiently, creating a circular close-loop biorefinery which required very little energy and materials to run and generated almost zero net waste while providing several major benefits. Future work should focus on extending the concept to other types of wastewaters, other species of macrophytes, and other species of earthworms. More precise process optimization and value addition should also be explored.

**Author Contributions:** Conceptualization, T.-A. and S.A.A.; methodology, T.-A., R.R. and S.A.A.; software, R.R. and T.-A.; validation, T.-A., R.R. and T.-A.; formal analysis, P.P. and T.-A.; investigation, P.P. and T.-A.; resources, S.A.A. and R.R.; data curation, P.P. and T.-A.; writing—original draft preparation, R.R. and T.-A.; writing—review and editing, P.P. and S.A.A.; visualization, T.-A.; supervision, S.A.A.; project administration, S.A.A. and T.-A.; funding acquisition, S.A.A. All authors have read and agreed to the published version of the manuscript.

**Funding:** This work is financed by Ministry of Water Resources, RD & GR, Government of India.

**Institutional Review Board Statement:** Not applicable.

**Informed Consent Statement:** Not applicable.

**Data Availability Statement:** All the pertinent data has been included in the paper.

**Acknowledgments:** Authors thank the Ministry of Water Resources, RD & GR, Government of India, for funding an R&D project that enabled the work and Department of Biotechnology, Government of India, for supporting the patenting of SHEFROL®.

**Conflicts of Interest:** The authors declare no conflict of interest.

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
