# Peer review of "A Circular Biorefinery-Integrating Wastewater Treatment with the Generation of an Energy Precursor and an Organic Fertilizer"

_sustainability, doi:10.3390/su14095714_

Round 1

Reviewer 1 Report

The manuscript is devoted to the creation of a wastewater treatment system with zero or almost zero discharge based on the cultivation of an aquatic plant and the production of useful products from excess biomass of it. The article is quite compact, well structured, clearly presented and interesting. I believe that it can be accepted for publication after addressing the minor comments that can be found in the pdf-file.

Author Response

Thank you for your reviewing, please check the attachment

Reviewer 2 Report

Great article and nice read. Below are my minor comments.

1) Usually, if you consider a WWT facility integrated with a biorefinery, the costs associated with WWT facility are very high due to high BOD/COD content produced from upstream deacetylation and hydrolysis processes. The wastewater needs to go through series of aerobic, anaerobic digestion, filtration, etc. steps to make sure the water released to a water body follows environmental regulatory limits. I would urge authors to include a section on economics of WWT facility and how it impacts the production cost at the biorefinery. 

2) What about the future directions of the study? Authors need to include this in the conclusions section.

3) I do not see any caveats included in the discussion section. For example, if some the variable assumptions were to be changed, how does it affect the overall productivity and yields?

Author Response

Thank you for your suggestions, please check the attachment.

Reviewer 3 Report

Dear Editor

The manuscript shows a solution to wastewater treatment. The results and discussion were based in "Taguchi plot".

Best regards!

Author Response

Thank you for your review.

Reviewer 4 Report

line 91: (Spillman ): what is this?

line 107: 40,000LPD of sewage went into 54,000LPD reactor. This flow is 74% of treatment capacity. You should explain the effect of this low flow.

Figure 1: Flow equalization-cum-sedimentation tank  should be drawn.

               Sampling places (2,4,6 and 8 hours) should be shown.

               Control channel should be drawn.

line 109: You delete 0.1m from 0.4m of depth as root depth. You should mention the density of roots; then, modify the depth based on the density.

line 164 - 173: You should mention 'your' materials and method in this study.

line 177: figure 3 in this place was not Figure 3.

line 181 : Table 1 must be Table 2.

line 188-189: You should mention effect of sedimentation tank.

line 189: `increase significantly`  is not correct. You should change the expression.

line 199: You should explain what 'gravitational energy' is.

line 209-210: You should show the data concerning 'peaking of the VFA yield may vary in the 4-8 day range.'

line 276: You should show 'national standards.'

Table 1: Concerning sodium hydroxide solution, 5.0(1) may be 5.0(2) and 7.5(1) may be 7.5(3). 

Author Response

Thank you for your suggestions, please check the attachment
